# Multiple vector-borne pathogens of domestic animals in Egypt

**Hend H. A. M. Abdullah**[1,2], **Nadia Amanzougaghene**[2], **Handi Dahmana**[2], **Meriem Louni**[2], **Didier Raoult**[2], **Oleg Mediannikov**[2]*

**1** Department of Parasitology and Animal Diseases, Veterinary Research Division, National Research Centre, Dokki, Giza, Egypt, **2** Aix Marseille Univ, IRD, AP-HM, MEPHI, IHU-Méditerranée Infection, Marseille, France

* olegusss1@gmail.com

**Data Availability Statement:** All relevant data are within the manuscript.

**Funding:** This study was supported by the Institut Hospitalo-Universitaire (IHU) Méditerranée

## Abstract

Vector Borne Diseases (VBDs) are considered emerging and re-emerging diseases that represent a global burden. The aim of this study was to explore and characterize vector-borne pathogens in different domestic animal hosts in Egypt. A total of 557 blood samples were collected from different animals using a convenience sampling strategy (203 dogs, 149 camels, 88 cattle, 26 buffaloes, 58 sheep and 33 goats). All samples were tested for multiple pathogens using quantitative PCR and standard PCR coupled with sequencing. We identified *Theileria annulata* and *Babesia bigemina* in cattle (15.9 and 1.1%, respectively), *T. ovis* in sheep and buffaloes (8.6 and 7.7%, respectively) and *Ba. canis* in dogs (0.5%) as well as *Anaplasma marginale* in cattle, sheep and camels (20.4, 3.4 and 0.7%, respectively) and *Coxiella burnetii* in sheep and goats (1.7 and 3%; respectively). New genotypes of *An. centrale*, *An. ovis*, *An. platys*-like and *Borrelia theileri* were found in cattle (1.1, 3.4, 3.4 and 3.4%, respectively), *An. platys*-like in buffaloes (7.7%), *An. marginale*, *An. ovis*, *An. platys*-like and *Bo. theileri* in sheep (3.4, 1.7, 1.7 and 3.4%, respectively), *An. platys*, *An. platys*-like and *Setaria digitata* in camels (0.7, 5.4 and 0.7%, respectively) and *Rickettsia africae*-like, *An. platys*, *Dirofilaria repens* and *Acanthocheilonema reconditum* in dogs (1.5, 3.4, 1 and 0.5%, respectively). Co-infections were found in cattle, sheep and dogs (5.7, 1.7, 0.5%, respectively). For the first time, we have demonstrated the presence of several vector-borne zoonoses in the blood of domestic animals in Egypt. Dogs and ruminants seem to play a significant role in the epidemiological cycle of VBDs.

## Author summary

Vector Borne Diseases (VBDs) are considered emerging and re-emerging diseases that represent a global burden. Diagnosis of these diseases is challenging due to nonspecific febrile illness, difficulty of isolation, and cross-reactivity of serological methods. Therefore, the current study is the first large-scale epidemiological study in which molecular screening and characterization of multiple vector-borne pathogens in different animal hosts were performed to better understand the endemicity of VBDs in Egypt. We detected for the first time *Anaplasma centrale*, *An. ovis*, a novel *An. platys*-like and *Borrelia theileri*

Infection, the National Research Agency under the program "Investissements d'avenir", reference ANR-10-IAHU-03, the Région Provence Alpes Côte d'Azur and European funding FEDER PRIMI. The authors acknowledge funding from the Science and Technology Development Fund (STDF) and Institut Francais d'Egypte (IFE) (ID: 30652) for the support of this research. The funders just supported the study through chemicals availability. The funders had no role in study design, data collection and analysis, decision to publish, or preparation of the manuscript.

**Competing interests:** The authors have declared that no competing interests exist.

in cattle, a new *An. platys*-like in buffaloes, *An. marginale*, *An. ovis*, a new *An. platys*-like and *Bo. theileri* in sheep, *An. platys*, a new *An. platys*-like and *Setaria digitata* in camels and *Rickettsia africae*-like, *An. platys*, *Dirofilaria repens* and *Acanthocheilonema reconditum* in dogs, in Egypt. These results imply that ruminants and dogs in Egypt are reservoirs for several neglected, emerging and re-emerging potentially new vector-borne pathogens that have significant implications in human health.

## Introduction

Vector Borne Diseases (VBDs) are emerging and re-emerging infectious diseases, that pose a health threat to humans, livestock, companion animals and wildlife [1]. VBDs are a global burden and cause severe economic losses through high mortality rates and production declines in the livestock industry, as well as impacts on human and animal health [2,3]. Moreover, about a quarter of vertebrate pathogens of veterinary importance are VBDs [4]. The World Organization for Animal Health (OIE) list includes many VBDs such as piroplasmoses, anaplasmoses and Q fever. The epidemiology and spread of VBDs are influenced by various factors such as globalization and increasing international trade, urbanization, climate change, travel and mobility of animals which pose unprecedented challenges to clinicians and veterinarians [5–6].

Piroplasmoses are tick-borne infectious diseases caused by apicomplexans of the order *Piroplasmida*, which includes three genera namely: *Theileria*, *Babesia* and *Cytauxzoon* [7]. *Theileria annulata*, *T. ovis* and *Babesia bigemina* are etiological agents of tropical theilerioses and babesiosis in ruminants especially cattle, buffalo and sheep [8]. Similarly, *Ba. canis* and *Ba. vogeli* are the main causative agents of canine babesiosis [9]. Piroplasmoses are common in Asia, Southern Europe and Africa [10]. The main clinical signs of piroplasmoses are fever and hemolytic anemia and deaths of up to 50% in the case of acute infection in susceptible herds [11,12]. Recovered animals may become asymptomatic carriers with long-term persistent infection [13,14]. Piroplasmoses have been detected in several provinces of Egypt and are widespread [15–18].

Anaplasmataceae include many tick-borne bacteria that infect mammals and consist of at least five genera: *Anaplasma*, *Ehrlichia*, *Neoehrlichia Neorickettsia*, and *Aegyptianella* [19–20]. Bovine anaplasmosis caused by *Anaplasma marginale* and *An. centrale* mainly in tropical and subtropical regions cause mild to severe anemia in ruminants [20,21]. Ovine anaplasmosis is a neglected mild disease in sheep, goats and wild ruminants caused by *An. ovis* and is common in different areas of the world [22,23]. In addition, there are many Anaplasmataceae bacteria pathogenic to dogs, such as *An. platys* and *Ehrlichia canis* [24,25]. Overall, these bacteria could cause persistent infection in mammals making them reservoir, which has lasting effect on the spread and new outbreaks of anaplasmosis [26,27]. In Egypt, anaplasmosis has been reported in cattle, water buffaloes and camels in different provinces [16,28–34].

Rickettsioses are bacterial infectious diseases that cause health problems in humans and animals worldwide [35,36]. Rickettsiae are divided into spotted fever group (SFG; mainly transmitted by ticks), typhus group (TG; transmitted by lice and fleas), *Rickettsia belli* group and *Rickettsia* (*R.*) *candensis* group [37]. *R. africae* is the most common rickettsial species in Africa that causes African tick-borne fever in humans [38]. Other rickettsiae such as *R. aeschlimannii*, *R. conorii* and *R. sibirica mongolitimonae*, *R. massiliae* have been detected in ticks and animals in Africa [39–43]. In Egypt, SFG have been identified in vectors, animals and humans since 1989 [44–48]. SFG rickettsiae were found in ticks (*Hyalomma* sp. and *Rhipicephalus*

*sanguineus*) collected in Sinai province [49–51]. Moreover, *R. siberica mongolitimonae* was detected in a French traveler returning from Egypt [52]. Finally, *R. africae* was detected by molecular biology in *Hyalomma* sp. and camels [53–55].

Borrelioses are zoonotic infectious diseases and are divided into two groups: Lyme disease group (caused by *Borrelia burgdorferi* and related species) and relapsing fever group [56]. Relapsing fever borrelioses are arthropod-borne spirochetal diseases, usually transmitted by soft ticks; they are common in subtropical regions worldwide [57]. In Africa, relapsing fever is most common in the northern hemisphere and is caused by various *Borrelia* spp. such as *Bo. hispanica*, *Bo. duttonii*, and *Bo. crocidurae* [57–60]. *Bo. theileri* is the etiological agent of bovine borreliosis in ruminants, which causes anemia and fever and, unlike other members of the relapsing fever spirochetes, is transmitted by hard ticks [58]. In Egypt, data on borrelioses in animal hosts are sparse. Only the few studies have detected *Bo. burgdorferi* [61,62] and *Bo. theileri* in hard ticks [62].

Q fever is a zoonosis that infects humans and animals through direct contact or a tick bite [63]. *Coxiella burnetii* is the causative agent of Q fever that may be severe in humans [64]. Infection in animals it is usually subclinical except that reproductive diminution and abortions may occur [65]. *Coxiella burnetii* infects a wide range of animals, especially sheep, goats, cattle and camels, which serve as reservoirs [64,66]. In Egypt, the seroprevalence of *C. burnetii* was estimated in buffaloes, sheep, cattle and camels [67–70]. In addition, *C. burnetii* has been detected molecularly in goats, camels and ticks (*H. dromedarii*) [70–72].

Filarial nematodes are vector-borne helminths belonging to the order Spiruridae, suborder Spirurina and families Filariidae and Onchocercidae and pose a serious threat to humans and livestock [73,74]. *Dirofilaria repens* and *D. immitis*, followed by *Acanthocheilonema* sp. are the most important etiological agents of filarial infections in dogs [9,73,75]. *Setaria digitata* is a filarial nematode of cattle and buffaloes and is not pathogenic to these natural hosts, but when transmitted by mosquitoes to accidental hosts such as camels and horses, it can have serious pathological effects [76,77]. In Egypt, information on filarial infections in ruminants and dogs are scarce. In Africa, there are some reports of filarial infections in different places of the continent [78–80].

Diagnosis of all these diseases is challenging due to the non-specific febrile illness, difficulty in isolation and cross reactivity of serological methods [35,59]. Therefore, the advanced molecular techniques have been used to increase the sensitivity and specificity of diagnosis, to detect previously unknown pathogens and distinguish closely related species [5]. In Egypt, the epidemiology and prevalence of these diseases remain neglected and poorly understood. To date, few studies have been conducted on individual VBDs in vectors or animal hosts. Here, we provide the first data for molecular screening and characterization of multiple vector-borne pathogens in different animal hosts to better understand the epidemiological approach of VBDs in Egypt.

## Materials and methods

### Ethical approval

This study was approved by the Medical Research Ethics Committee at the National Research Centre, Egypt with the number 19058.

### Study area and samples collection

We conducted a cross-sectional observational study with a total of 557 apparently healthy domestic animals (203 dogs, 149 camels, 88 cattle, 26 buffaloes, 58 sheep and 33 goats) using a convenience sampling strategy [81]. Animal blood samples were randomly collected from

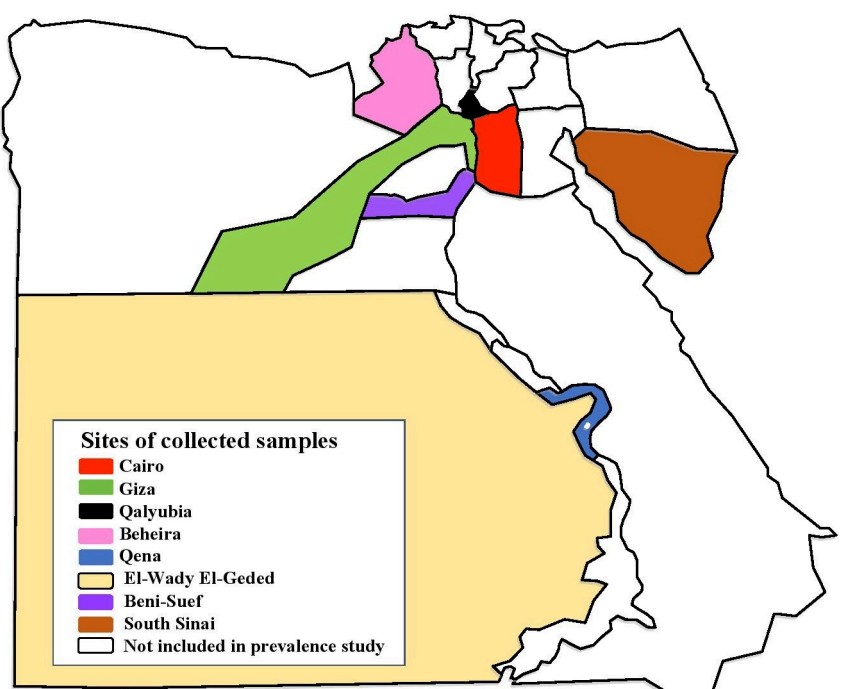

**Fig 1. Map of Egypt showing the different provinces where the blood samples from different animal hosts were collected for our study.** https://en.wikipedia.org/wiki/Governorates_of_Egypt and the picture has CC BY-SA 3.0.

different provinces in Egypt between 2016 and 2018. The details of the sample locations were presented in Fig 1 and Table 1. For each animal host, 5 ml of blood was collected in a sterile EDTA tube using a sterile syringe and stored at -20˚C for molecular purposes. The prevalence of infection of different pathogens by different animal hosts was calculated according to Thrusfield et al. [81].

## DNA extraction

DNA was extracted from 200 μl of each blood sample using EZ1 DNA Blood Kit (Qiagen, Hilden, Germany) according to the manufacturer's instructions. The extracted DNA was stored at -20˚C until use for molecular screening.

## Screening of multiple pathogen DNA by qPCR

All samples were first screened for pathogen DNA by qPCR using genus-specific primers and probes targeting the 5.8S rRNA gene of piroplasms, the 23S rRNA gene of Anaplasmataceae, the *gltA* gene of *Rickettsia* sp., the 16S rRNA gene of *Borrelia* sp., the IS1111 of *C. burnettii*, BartoITS3 of *Bartonella* sp. and the pan-fil 28S rRNA gene of Filariidae. For positive Filariidae in dog samples, a triplex qPCR targeting Cox1 was used to detect *D. immitis*, *D. repenes* and *Ac. reconditum*. The sequence of primers and probes used in this study is showed in Table 2. The qPCR was preformed using a CFX 96 Real Time System (Bio-Rad Laboratories, Foster City, CA, USA). The total reaction volume of 20 μl included 10 μl of Eurogentec Master Mix Roche, 0.5 μl of each primer, 0.5 μl of FAM-labeled probe, 0.5 μl of UDG, 5 μl of DNA template, and 3 μl of DNAse- and RNAse-free water. Thermal cycling was performed according to the instructions provided by the manufacturer of the Master Mix PCR kit. To evaluate the PCR reaction, a positive control (pathogen DNA) and a negative control were added to each

**Table 1. The information data of collected samples.**

| Provinces | Geographic coordinates | Animal Hosts | Locations | Numbers of Animals |
|---|---|---|---|---|
| **Cairo** | 30° 03' 45.47" N, 31° 14' 58.81" E | Dog | Police Academy (El-Abbasia) | 75 |
| | | | Police Academy (El-Tagamoa) | 67 |
| | | | Police Academy (El-Dowaika) | 61 |
| | | Camel | Police Academy (Gasr-El Swiss) | 52 |
| **Giza** | 29° 58' 27.00" N, 31° 08' 2.21" E | Camel | Police Academy (El-Haram) | 96 |
| | | sheep | households | 5 |
| | | Goat | households | 6 |
| **Beni-Suef** | 29° 03' 60.00" N, 31° 04' 60.00" E | Cattle | households | 63 |
| | | Sheep | households | 48 |
| | | Goat | households | 20 |
| | | Buffalo | households | 20 |
| **Qalyubia** | 30.41°N, 31.21°E | Cattle | households | 2 |
| | | Buffalo | households | 6 |
| | | Goat | households | 2 |
| **Sinai** | 28° 32' 13.79" N, 33° 58' 14.39" E | Sheep | households | 5 |
| | | Goat | households | 5 |
| | | Camel | Free rearing | 1 |
| **El-Wady El-Geded** | 24°32'44"N, 27°10'24"E | Cattle | households | 11 |
| **Qena** | 26° 09' 60.00" N, 32° 42' 59.99" E | Cattle | households | 10 |
| **Beheira** | 30.61°N, 30.43°E | Cattle | households | 2 |

reaction. The sample was considered positive if the cycle threshold (Ct) was less than 35 Ct [82].

## Standard PCR and sequencing

All samples considered positive by qPCR were subjected to standard PCR and sequencing. Primers targeting 969 bp and 1200 bp region of the 16S rRNA gene, respectively, were used to identify *Piroplasma* and *Borrelia*. For the identification of Anaplasmataceae, standard PCR were performed with primers targeting a 520 bp fragment of the 23S rRNA gene. The positive samples with 23S rRNA gene were confirmed with *Anaplasma* genus-specific primers targeting the 525 bp fragment of the *rpoB* gene. *Rickettsia* genus-specific primers targeting the *gltA* gene were used and the positive samples were confirmed by the *ompB* gene. Moreover, multispacer typing (MST) for *C. burnetii* was performed by amplifying of three intergenic spacers (Cox2, Cox5 and Cox18). Identification of Filariidae was performed using 18S rRNA primers targeting 1155 bp. All primer sequences used in standard PCR and sequencing are listed in Table 2. All PCR reactions were performed in an Applied Biosystems 2720 Thermal Cycler model (Thermo Fisher Scientific Courtaboef, France) using AmpliTaq 360 Master Mix (Thermo Fisher Scientific Courtaboef, France) according to the manufacturer's recommendations. Negative and positive controls were included in each reaction. PCR products were visualized by electrophoresis on a 1.5% agarose gel stained with Syper Safe stain (Invitrogen, USA) and analyzed using Lab Image software (BioRad, Marnes-La-Coquette, France).

PCR products were purified using NucleoFast 96 PCR plates (Macherey Nagel, EURL, Hoerdt, France), according to the manufacturer's recommendation. The purified PCR products were sequenced using the Big Dye Terminator Cycle Sequencing Kit (Perkin Elmer Applied Biosystems, Foster City, CA, USA) with an ABI automated sequencer (Applied Biosystems). The sequences obtained were assembled and edited using ChromasPro software

**Table 2. Primers and probes used for qPCR, Standard PCR and sequencing in this study.**

| Microorganisms | Targeted gene | Primers F, R (5'-3') and Probes S (6FAM–TAMRA) | Tm | References |
|---|---|---|---|---|
| Piroplasmida | 5.8S rRNA | 5.8S-F5-AYYKTYAGCGRTGGATGTC | - | [135] |
| | | 5.8S-R-TCGCAGRAGTCTKCAAGTC | | |
| | | 5.8S-S-TTYGCTGCGTCCTTCATCGTTGT | | |
| | 18S rRNA | piro18SF1- GCGAATGGCTCATTAIAACA | 58˚C | |
| | | piro18SF4-TTTCAGMCTTGCGACCATACT | | |
| | | piro18SF3-GTAGGGTATTGGCCTACCG | | |
| | | piro18SR3-AGGACTACGACGGTATCTGA | | |
| Anaplasmataceae | 23S rRNA (TtAna) | TtAna-F-TGACAGCGTACCTTTTGCAT | - | [24] |
| | | TtAna-R-GTAACAGGTTCGGTCCTCCA | | |
| | | TtAna-S-CTTGGTTTCGGGTCTAATCC | | |
| | 23S rRNA | Ana23S-212F-GTTGAAAARACTGATGGTATGCA | 55˚C | |
| | | Ana23S-753R-TGCAAAAGGTACGCTGTCAC | | |
| | Ana-rpoB | rpoB-F-GCTGTTCCTAGGCTYTCTTCGCGA | 52˚C | |
| | | rpoB-R-AATCRAGCCAVGAGCCCCTRTAWGG | | |
| Rickettsia sp. | gltA (RKNDO3) | RKNDO3-F-GTGAATGAAAGATTACACTATTTAT | - | [136] |
| | | RKNDO3-R-GTATCTTAGCAATCATTCTAATAGC | | |
| | | RKNDO3-S-CTATTATGCTTGCGGCTGTCGGTTC | | |
| | gltA | CS2D-ATGACCAATGAAAATAATAAT | 50˚C | [137] |
| | | CSEnd-CTTATACTCTCTATGTACA | | |
| | OmpB | 120-M59-CCGCAGGGTTGGTAACTGC | 50˚C | [138] |
| | | 120-607-AATATCGGTGACGGTCAAGG | | |
| | | 120-1497- CCTATATCGCCGGTAATT | | |
| Borrelia sp. | Internal transcribed spacer 16S RNA (Bor ITS4) | BorITS4-F-GGCTTCGGGTCTACCACATCTA | - | [139] |
| | | BorITS4-R-CCGGGAGGGGAGTGAAATAG | | |
| | | BorITS4-S-TGCAAAAGGCACGCCATCACC | | |
| | 16S rRNA | 16S-F-GCTGGCAGTGCGTCTTAAGC | 57˚C | |
| | | 16S-R-GCTTCGGGTATCCTCAACTC | | |
| Coxiella burnetii | Insertion Sequence (IS1111) | IS1111-F-CAAGAAACGTATCGCTGTGGC | - | [63] |
| | | IS1111-R-CACAGAGCCACCGTATGAATC | | |
| | | IS1111-S-CCGAGTTCGAAACAATGAGGGCTG | | [140] |
| | Cox2 | Cox2-F-CAACCCTGAATACCCAAGGA | 57˚C | |
| | | Cox2-R-GAAGCTTCTGATATAGGCGGGA | | |
| | Cox5 | Cox5-F-CAGGAGCAAGCTTGAATGCG | | |
| | | Cox5-R-TGGTATGACAACCCGTCATG | | |
| | Cox18 | Cox18-F-CGCAGACGAATTAGCCAATC | | |
| | | Cox18-R-TTCGATGATCCGATGGCCTT | | |
| Bartonella sp. | Internal transcribed spacer16S (BartoITS3) | BartoIRS3-F-GATGCCGGGGAAGGTTTTC | - | [141] |
| | | BartoIRS3-R-GCCTGGGAGGACTTGAACCT | | |
| | | BartoIRS3-S-GCGCGCGCTTGATAAGCGTG | | |
| Filariidae | Pan-fil 28S rRNA | qFil-28S-F-TTGTTTGAGATTGCAGCCCA | - | [142] |
| | | qFil-28S-R-GTTTCCATCTCAGCGGTTTC | | |
| | | qFil-28S-S-CAAGTACCGTGAGGGAAAGT | | |
| | Triplex TaqMan Cox1 | Fil.COI.749-F-CATCCTGAGGTTTATGTTATTATTTT | | [143] |
| | | D.imm.COI.777-S-CGGTGTTTGGGATTGTTAGTG | | |
| | | D.rep.COI.871-S-TGCTGTTTTAGGTACTTCTGTTTGAG | | |
| | SSU rRNA (18S) | Fwd.18S.631-TCGTCATTGCTGCGGTTAAA | 55˚C | [144] |
| | | Rwd.1465-GGTTCAAGCCACTGCGATTAA | | |

(ChromasPro 1.7, Technelysium Pty Ltd., Tewantin, Australia), and the corrected sequences were compared with the sequences available in GenBank by BLAST (https://blast.ncbi.nlm.nih.gov/Blast.cgi).

## Phylogenetic analyses

Multiple sequence alignments were performed between the obtained sequences and other reference sequences in GenBank using CLASTAL W in MEGA software version X [83]. Phylogenetic trees were inferred using the Maximum-Likelihood method and Tamura-Nei model with 500 bootstrap replicates in MEGA X software [83,84].

## Results

In this study, all samples (557) were screened by qPCR. None of the animals were positive for *Bartonella* sp., while different animal hosts were positive for piroplasms, *Anaplasma* sp., *Rickettsia* sp., *Borrelia* sp., *C. burnettii* and *Filaria* sp (Table 3).

Fifty of 557 (8.9%) animal hosts were positive for piroplasms based on 5.8S rRNA qPCR system. Standard PCR and sequencing based on 18S rRNA gene succeeded in amplifying and identifying two *Theileria* sp.; *T. annulata* in cattle (14/88), *T. ovis* in sheep and buffaloes (5/58 and 2/26, respectively) and two *Babesia* sp.; *Ba. bigemina* in cattle (1/88) and *Ba. canis* in dogs

**Table 3. The prevalence of pathogens in animals by PCR.**

| Animal Hosts | No. of examined Animals (Total = 557) | Pathogens amplified | No. of infected Animals (%) |
|---|---|---|---|
| **Cattle** | 88 | **Piroplasmida** | **15/88 (17%)** |
| | | *T. annulata* | 14/88 (15.9%) |
| | | *Ba. bigemina* | 1/88 (1.1%) |
| | | **Anaplasmataceae** | **25/88 (28.4%)** |
| | | *An. marginale* | 18/88 (20.4%) |
| | | *An. centrale* | 1/88 (1.1%) |
| | | *An. ovis* | 3/88 (3.4%) |
| | | *An.* platys-like | 3/88 (3.4%) |
| | | ***Borrelia* sp.** | **3/88 (3.4%)** |
| | | *Bo. theileri* | |
| | | **Co-infection :** | **5/88 (5.7%)** |
| | | *An. marginale + T. annulata* | 2/88 (2.3%) |
| | | *An. marginale + Bo. theileri* | 1/88 (1.1%) |
| | | *An. centrale + T. annulata* | 1/88 (1.1%) |
| | | *An.* platys-like *+ Ba. bigemina* | 1/88 (1.1%) |
| **Buffalo** | 26 | **Piroplasmida** | **2/26 (7.7%)** |
| | | *T. ovis* | |
| | | **Anaplasmataceae** | **2/26 (7.7%)** |
| | | *An.* platys-like | |
| **Sheep** | 58 | **Piroplasmida** | **5/58 (8.6%)** |
| | | *T. ovis* | |
| | | **Anaplasmataceae** | **4/58 (6.9%)** |
| | | *An. marginale* | 2/58 (3.4%) |
| | | *An. ovis* | 1/58 (1.7%) |
| | | *An.* platys-like | 1/58 (1.7%) |
| | | ***Borrelia* sp.** | **2/58 (3.4%)** |
| | | *Bo. Theileri* | |
| | | ***Coxiella burnetii*** | **1/58 (1.7%)** |
| | | **Co-infection :** | **1/58 (1.7%)** |
| | | *An.* platys-like *+ Bo. theileri* | |
| **Goat** | 33 | ***Coxiella burnetii*** | **1/33 (3%)** |
| **Camel** | 149 | **Anaplasmataceae** | **10/149 (6.7%)** |
| | | *An. marginale* | 1/149 (0.7%) |
| | | *An.* platys | 1/149 (0.7%) |
| | | *An.* platys-like | 8/149 (5.4%) |
| | | **Filariidae** | **1/149 (0.7%)** |
| | | *S. digitate* | |
| **Dog** | 203 | **Piroplasmida** | **1/203 (0.5%)** |
| | | *Ba. canis* | |
| | | **Anaplasmataceae** | **7/203 (3.4%)** |
| | | *An.* platys | |
| | | ***Rickettsia* sp.** | **3/203 (1.5%)** |
| | | *Rickettsia africae*-like | |
| | | **Filariidae** | **3/203 (1.5%)** |
| | | *D. repens* | 2/203 (1%) |
| | | *Ac. reconditum* | 1/203 (0.5%) |
| | | **Co-infection :** | **1/203 (0.5%)** |
| | | *R. africae*-like *+ Anaplasma* | |

(1/203). However, camels and goats were free of Piroplasmida DNA. The overall prevalence of piroplasmoses in different animal hosts was 23/557 (4.1%) as it was 17% in cattle, 8.6% in sheep, 7.7% in buffaloes and 0.5% in dogs. In our study, BLAST analysis revealed that cattle were positive for *T. annulata* and *Ba. bigemina*, including two genotypes of *T. annulata*, one genotype in 13 cattle with 100% (910/910) similarity to those of *T. annulata* detected in donkey blood in Turkey (GenBank: MG569892), a new genotype in one cattle with 99% (908/910) identity to the same reference dataset, and a new genotype of *Ba. bigemina* in one cattle with 99% (865/866) identity to those of *Ba. bigemina* detected in cattle blood from Switzerland (GenBank: KM046917). Similarly, we found that 5 sheep and 2 buffaloes were positive for a genotype of *T. ovis* with 100% (897/897) identity to *T. ovis* detected in wild sheep from Turkey (GenBank: KT851427). Finally, we identified *Ba. canis* in a dog with 100% (884/884) similarity to those of *Ba. canis vogeli* detected in a dog from Egypt (GenBank: AY371197). The phylogenetic tree of these genotypes was illustrated in Fig 2.

For Anaplasmataceae, 172 out of 557 (30.9%) animal hosts were positive for anaplasmoses by 23S rRNA qPCR system. Based on the 23S rRNA gene, only 87 out of 557 animal hosts were successfully amplified by standard PCR, consequently, sequencing identified only 48 out of 557. The overall prevalence of anaplasmoses in different animals was 8.6%, with 28.4% in cattle (25/88), 6.9% in buffaloes (4/58), 7.7% in sheep (2/26), 6.7% in camels (10/149) and 3.4% in dogs (7/203), while goats were free of *Anaplasma* DNA. BLAST analysis revealed that cattle, sheep and camel were positive *An. marginale*, including two different genotypes of *An. marginale*, the first originated from sixteen cattle, two sheep and one camel with 100% (455/455) similarity to those of *An. marginale* detected in *Rh. bursa* collected from cattle in France (GenBank KY498335), and another new genotype was detected in two cattle with 99% (454/455) identity to the same reference dataset (GenBank KY498335). Moreover, one case of cattle was positive for *An. centrale* with 100% identical to *An. centrale* strain Israel (GenBank NR076686). From cattle and sheep, a genotype of *An. ovis* was identified with 100% (454/454) similarity to *An. ovis* in sheep blood from Niger (GenBank KY644694). We found that dogs and camels were positive for *An. platys*, including two different genotypes of *An. platys*, one genotype from six dogs and one camel with 100% (458/458) identity to *An. platys* in dog blood from France (GenBank KM021425) and another genotype from one dog with 100% (458/458) homology to *An. platys* in dog blood from France (GenBank KM021414). Finally, from cattle, buffaloes, sheep and camels, a new potential *Anaplasma* sp. was identified including, four different genotypes of this *Anaplasma* sp., the first genotype from six camels, the second from two camels, the third from one cattle and one sheep and the last from two cattle and two buffaloes with 98% (450/458), 98% (448/458), 98% (447/458) and 97% (446/458) similarity, respectively, to *An. platys* in dog blood from France (GenBank KM021414). Sequence analysis of this *Anaplasma* species revealed that this species has a homology score below 99% (more than 10 nucleotides different) and are closely related to *An. platys*, that means these sequences could be considered as potential new species of *Anaplasma* and can be called as *An. platys*-like. The phylogenetic tree showed that the new potential *Anaplasma* sp. in two separates and well-supported branches (bootstraps 99 and 96) belong to the cluster of *An. platys* (Fig 3).

To better characterize different *Anaplasma* genotypes, *rpoB* genus-specific PCR primers were applied and 23 good quality sequences were identified. The result revealed that, 12 cattle and one sheep were positive for a genotype of *An. marginale* with 100% (487/487) homology with *An. marginale* in *Rhipicephalus bursa* from France (KY498345), and another genotype of *An. marginale* from one cattle with 99% (486/487) similarity with the same reference dataset. We also identified that cattle and sheep were positive for *An. ovis*, one genotype was found in two cattle and another in a sheep with 100% (489/489) and 99% (487/489) identical to those of *An. ovis* in sheep blood from Niger (GenBank KY644695), respectively. From dogs, we

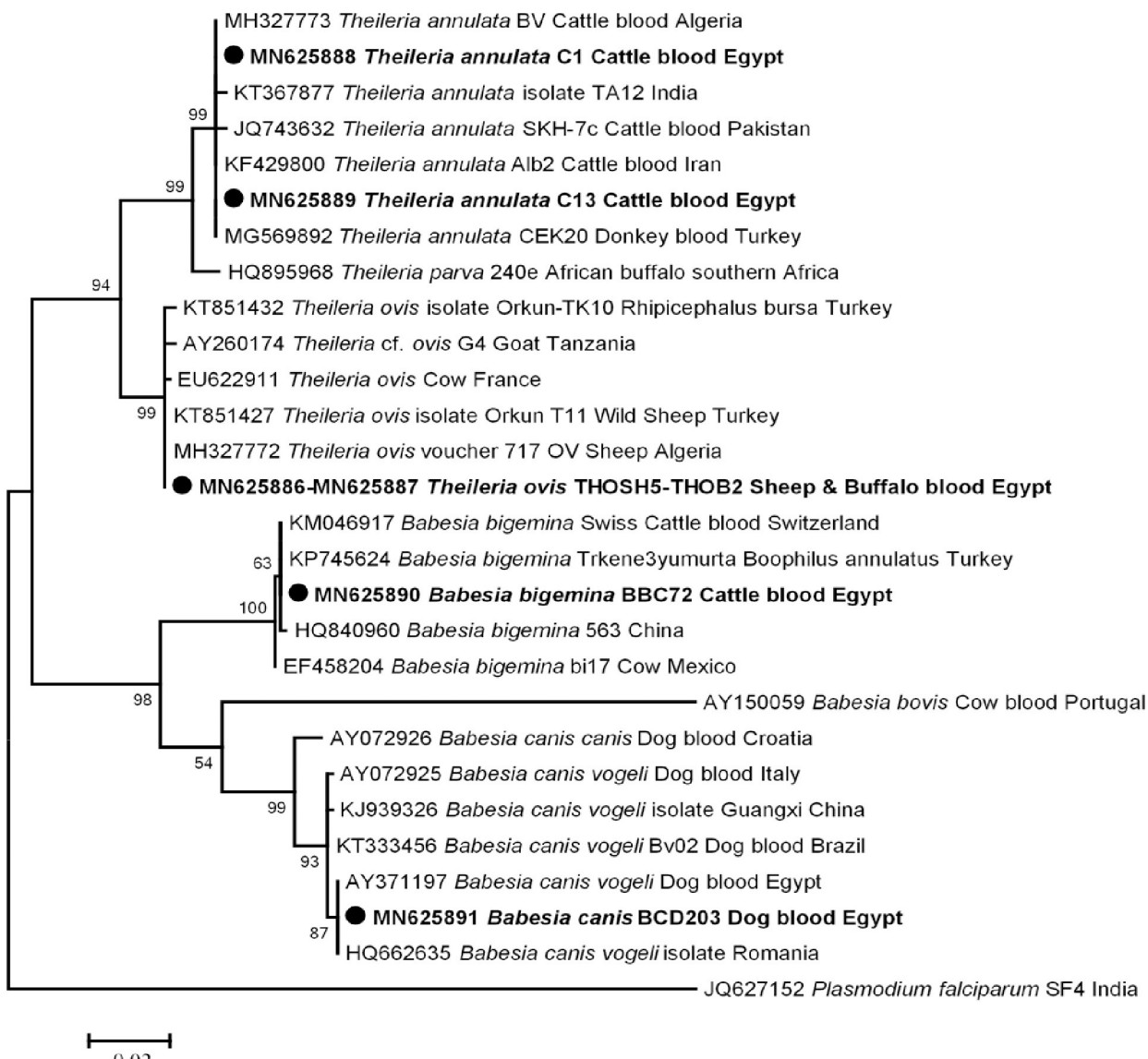

**Fig 2. 18S rRNA based phylogenetic analysis of genotypes identified in this study.** Phylogenetic tree highlighting the position of *Theileria* sp. and *Babesia* sp. in the present study (**Bold**) related to other *Theileria* sp. and *Babesia* sp. available in GenBank. The sequence of 18S rRNA were aligned using CLUSTAL W and phylogenetic inferences were constructed in MEGA X using Maximum Likelihood based on Tamura-Nei Model for nucleotide sequences with 500 bootstrap replicates. There was a total of 1066 positions in the final dataset. The scale bar represents a 2% nucleotide sequence divergence.

identified a new genotype of *An. platys* obtained from two dogs with 99% (488/489) homology to *An. platys* in dog blood from France (GenBank KX155493). Finally, from cattle, buffaloes and sheep, a new potential species of *Anaplasma*. was identified, its sequences had a homology score of less than 90%, confirming that these sequences are likely to be a new potential species of *Anaplasma* (like 23S rRNA gene). The only two different genotypes (one from two buffaloes and another from a cattle and a sheep) showed a low identity of 89% (432/486) and 88% (431/486), respectively, with *An. platys* in dog blood from France (GenBank KX155493), while identification of the genotype derived from camels failed. Phylogenetic analysis revealed a new potential *Anaplasma* sp. (*An. platys*-like) in a separate and well-supported branch (bootstraps 99) with the same clade belonging to *An. platys* (Fig 4).

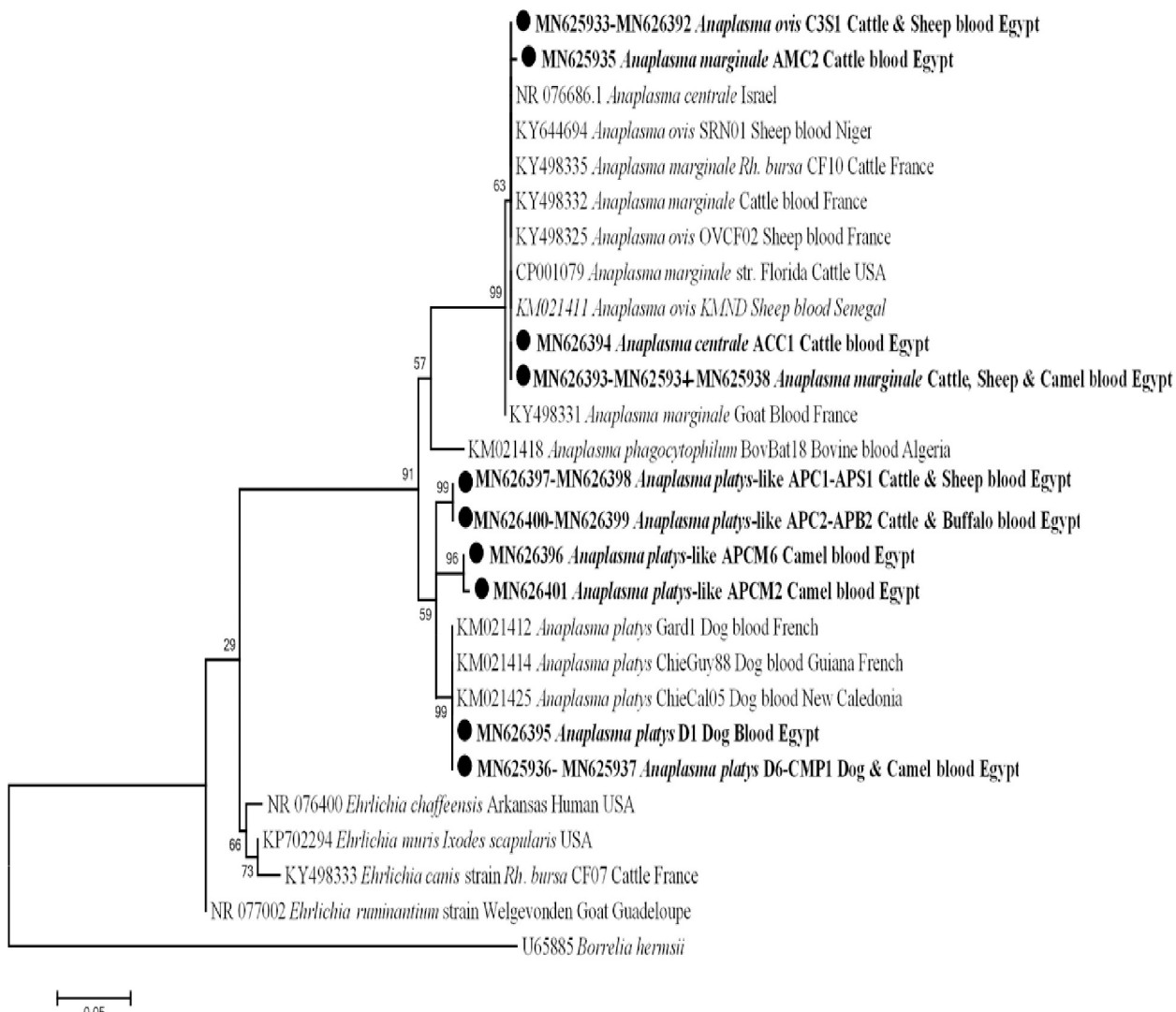

**Fig 3. 23S rRNA based phylogenetic analysis of genotypes identified in this study.** Phylogenetic tree highlighting the position of *Anaplasma* sp. in the present study (Bold) related to other *Anaplasma* sp. and *Ehrlichia* sp. available in GenBank. The sequence of 23S rRNA were aligned using CLUSTAL W and phylogenetic inferences were constructed in MEGA X using Maximum Likelihood based on Tamura-Nei Model for nucleotide sequences with 500 bootstrap replicates. There was a total of 432 positions in the final dataset. The scale bar represents a 5% nucleotide sequence divergence.

Rickettsial infection was detected by qPCR targeting *gltA* gene in dogs (3/557; 0.54%); the other animal hosts were free of rickettsiosis. To identify *Rickettsia* sp., standard PCR and sequencing were performed using *gltA* gene, and it was possible to amplify a 728 bp fragment of this gene from these three positive samples. A BLAST search of the obtained sequences with those in GenBank revealed that two different genotypes, one genotype was 100% (728/728) identical with *R. africae* previously detected in *H. dromedarii* from Egypt (GenBank: HQ335126), and the other sequence had 99% (726/728) identity with the same reference. Moreover, *ompB* gene was used to confirm the detection of *R. africae*-like infection in dogs. Based on the BLAST search, the sequences obtained from dogs were identified as *R. africae* (GenBank: MN629894) and showed (757/758) 99% similarity with the reference stain of *R. africae* detected in a traveler returning from Tanzania (GenBank: KU721071). The phylogenetic tree of these *R. africae*-like in dogs based on *gltA* was shown in Fig 5.

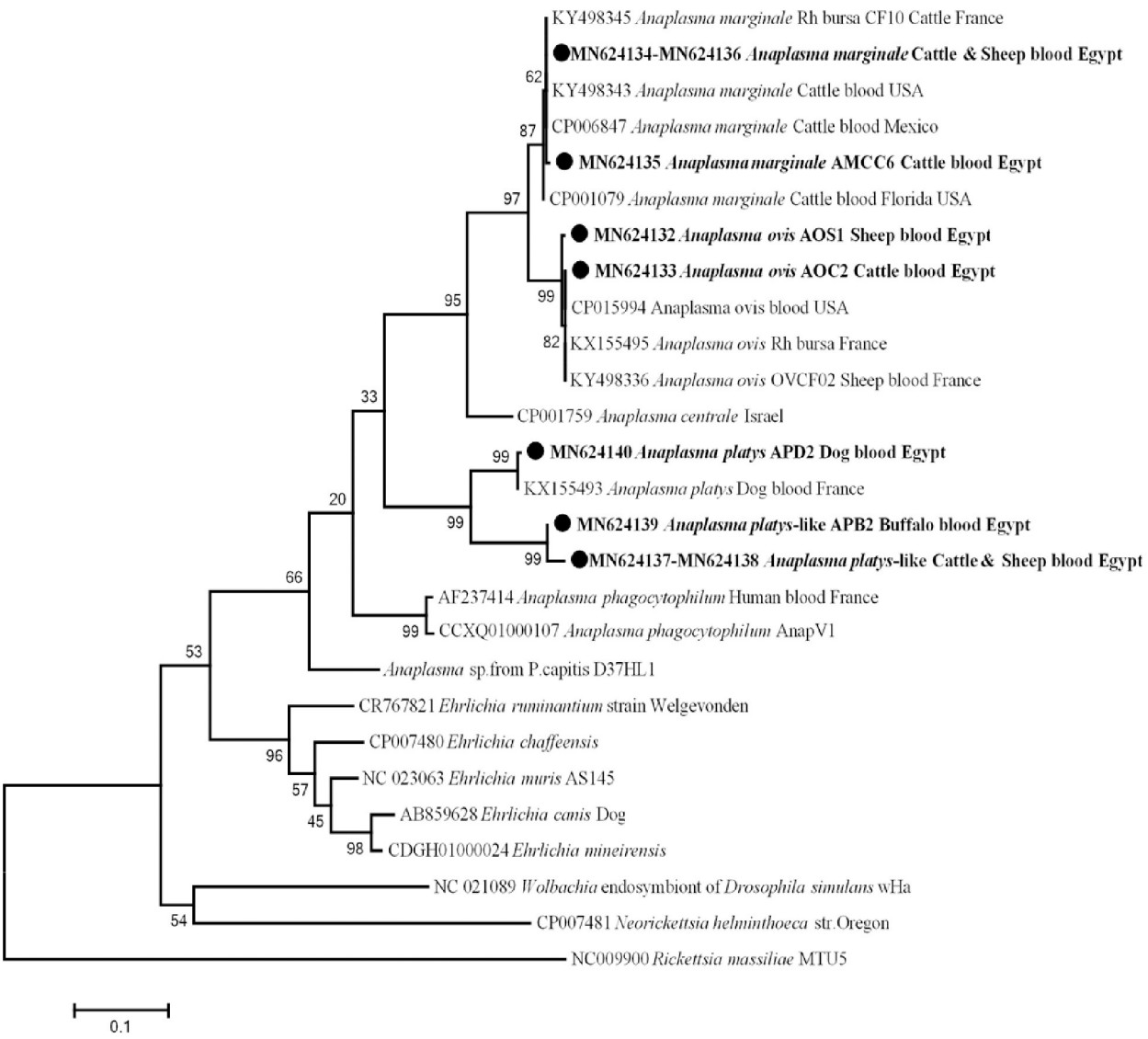

**Fig 4. *rpoB* gene based phylogenetic analysis of genotypes identified in this study.** Phylogenetic tree highlighting the position of *Anaplasma* sp. in the present study (**Bold**) related to other *Anaplasma* sp. and *Ehrichia* sp. available in GenBank. The sequence of *rpoB* gene were aligned using CLUSTAL W and phylogenetic inferences were constructed in MEGA X using Maximum Likelihood based on Tamura-Nei Model for nucleotide sequences with 500 bootstrap replicates. There was a total of 534 positions in the final dataset. The scale bar represents a 10% nucleotide sequence divergence.

Screening of *Borrelia* sp. in all animal hosts we found that 3 cattle and 2 sheep were positive for *Borrelia* sp. (5/557; 0.9%). Standard PCR and sequencing using 16S rRNA gene identified it as *Bo. theileri*. Alignment of five obtained sequences of *Borrelia* sp. from our samples revealed that all sequences were identical to each other. Furthermore, comparison of the obtained sequences with sequences from the GenBank database showed that 1139/1143 (99%) identity with *Bo. theileri* detected in *Rh. geigyi* in Mali (GenBank: KF569941). The phylogenetic position of this new *Bo. theileri* genotype was shown in Fig 6.

Two out of 557 (0.36%) blood samples from one sheep and one goat tested positive for *C. burnetii* DNA by qPCR targeting IS1111. MST genotyping was performed using Cox2, Cox5 and Cox18, with only Cox2 successfully identified and the other spacers failing amplification.

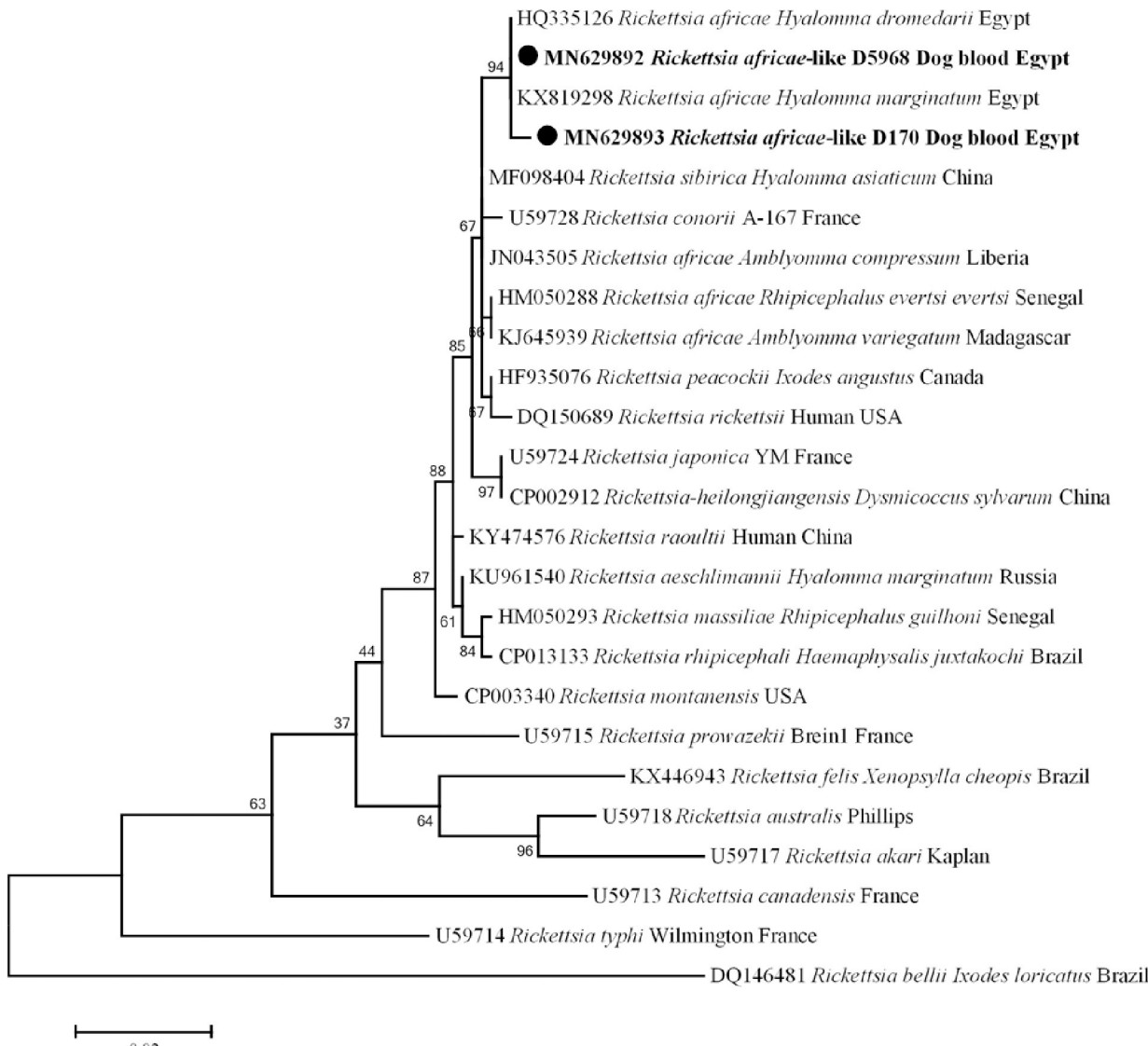

**Fig 5. *gltA* gene based phylogenetic analysis of genotypes identified in this study.** Phylogenetic tree highlighting the position of *Rickettsia* sp. in the present study (**Bold**) related to other *Rickettsia* sp. available in GenBank. The sequence of *gltA* gene were aligned using CLUSTAL W and phylogenetic inferences were constructed in MEGA X using Maximum Likelihood based on Tamura-Nei Model for nucleotide sequences with 500 bootstrap replicates. There was a total of 728 positions in the final dataset. The scale bar represents a 2% nucleotide sequence divergence.

A BLAST search for the two sequences obtained showed that (351/351) 100% identity with the reference sequences of *C. burnetii* recorded in GenBank.

Concerning Filariidae, four out of 557 (0.7%) animal hosts collected from three dogs and one camel tested positive for *Filaria* sp. DNA. By BLAST analyses, two dogs were found to have *D. repens* with 100% identity to those of *D. repens* previously detected in a Japanese woman returned from Europe (GenBank AB973229), and another sequence obtained from one dog showed 99% (1114/1119) similarity to *Ac. viteae* (GenBank: DQ094171). Moreover, *S. digitata* with (1107/1111) 99% identity to *S. digitata* from UK (GenBank: DQ094175) was found in a camel. The phylogenetic analysis of these *Filaria* sp. was constructed and presented in Fig 7.

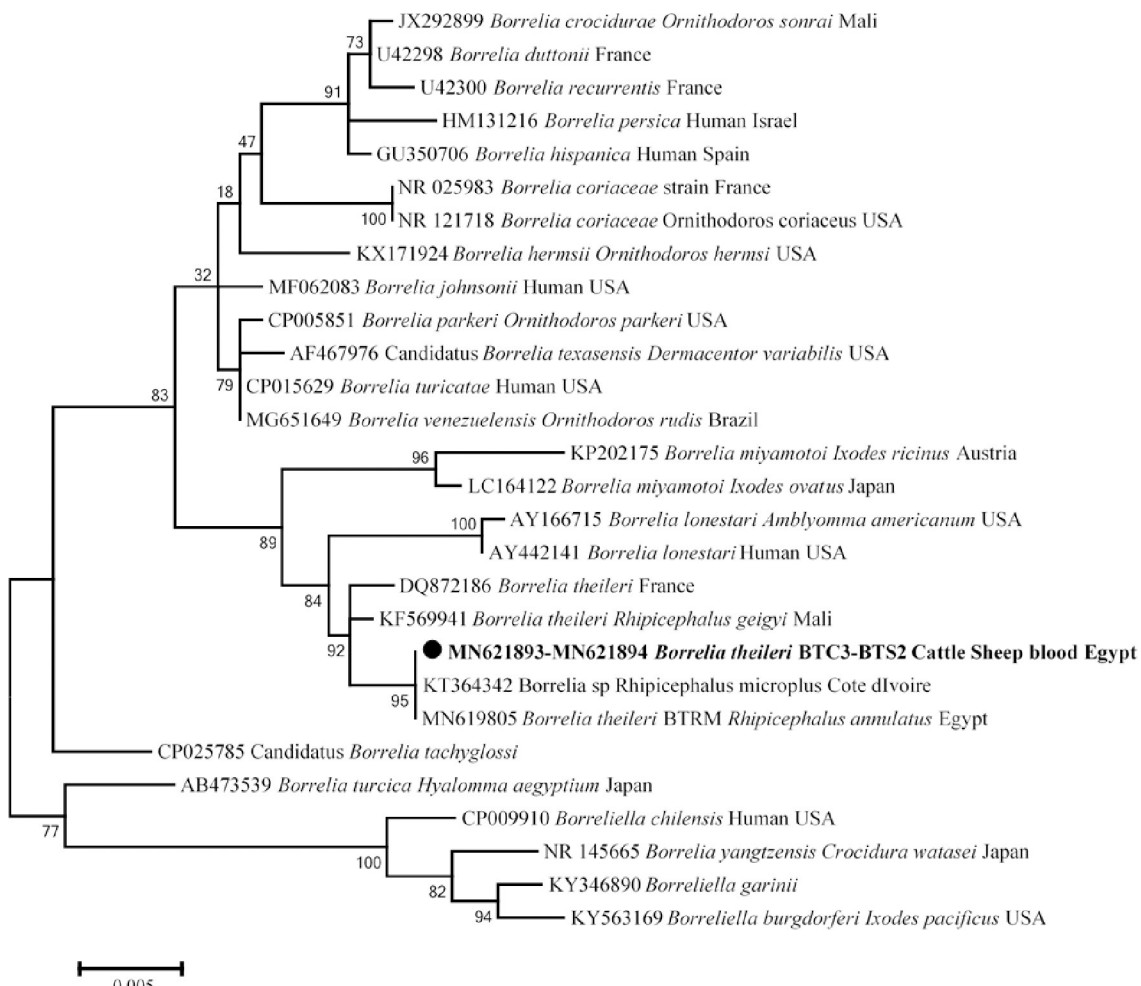

**Fig 6. 16S rRNA based phylogenetic analysis of genotypes identified in this study.** Phylogenetic tree highlighting the position of *Borrelia* sp. in the present study (**Bold**) related to other *Borrelia* sp. available in GenBank. The sequence of 16S rRNA were aligned using CLUSTAL W and phylogenetic inferences were constructed in MEGA X using Maximum Likelihood based on Tamura-Nei Model for nucleotide sequences with 500 bootstrap replicates. There was a total of 1142 positions in the final dataset. The scale bar represents a 5% nucleotide sequence divergence.

Finally, seven of different animal hosts were positive for more than one vector-borne pathogen (co-infections; 7/557; 1.3%). In cattle, five co-infections were observed (5/88; 5.7%) as *An. marginale* plus *T. annulata* (2/88; 2.3%), *An. marginale* plus *Bo. theilerii* (1/88; 1.1%), *An. centrale* plus *T. annulata* (1/88; 1.1%) and *An. platys*-like with *Ba. bigemina* (1/88; 1.1%). Moreover, one co-infection in sheep was recorded as *An. platys*-like plus *Bo. theilerii* (1/58; 1.7%) and one case in dogs *R. africae*-like with *Anaplasma* (1/203; 0.5%) (Table 3).

## Discussion

The sustainable and economic progress of developing countries depends mainly on domestic animal resources, as they provide vital food, draught power and manure for crop production, and generate income [85]. However, animal-associated diseases, especially, VBDs are a global burden [2]. Recently, the spectrum of VBDs affecting animals has expanded and the attention of clinicians and veterinarians is growing. Therefore, the diagnosis of VBDs is crucial to

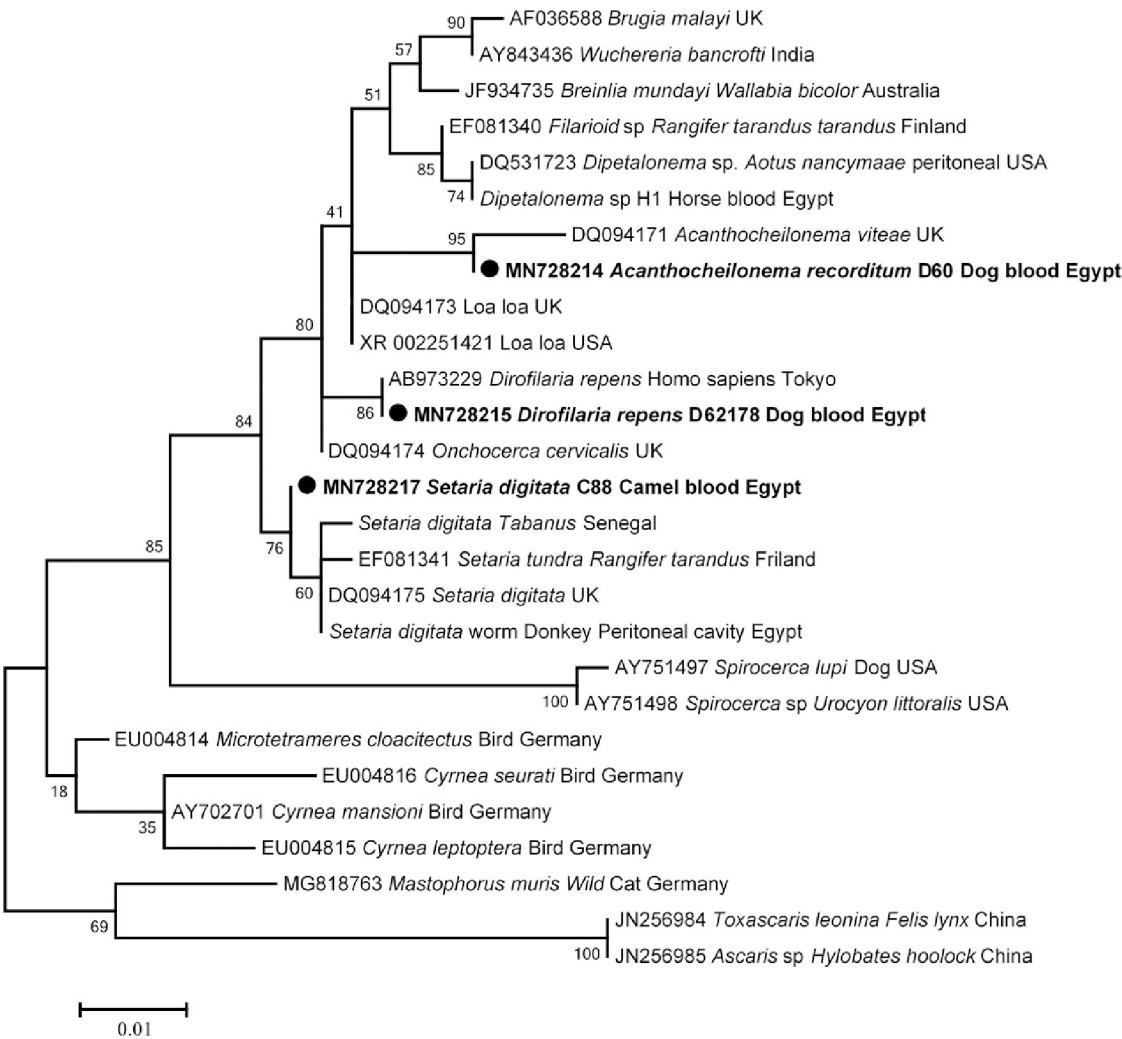

**Fig 7. 18S rRNA based phylogenetic analysis of genotypes identified in this study.** Phylogenetic tree highlighting the position of *Filaria* sp. in the present study (**Bold**) related to other *Filaria* sp. available in GenBank. The sequence of 18S rRNA were aligned using CLUSTAL W and phylogenetic inferences were constructed in MEGA X using Maximum Likelihood based on Tamura-Nei Model for nucleotide sequences with 500 bootstrap replicates. There was a total of 1110 positions in the final dataset. The scale bar represents a 10% nucleotide sequence divergence.

develop the epidemiological mapping of these diseases and this can be achieved through the advances in molecular biology [86].

Concerning piroplasmoses, the overall prevalence among animal hosts was 4.1%, including the highest prevalence among cattle 17%, then sheep 8.6%, buffaloes 7.7% and dogs 0.5%. Based on the 18S rRNA gene, two genotypes of *T. annulata* was detected in cattle from different provinces (El-Wady El-Geded, Beni-Suef, Qena and Beheira) and one case of *Ba. bigemina* was detected in cattle from Beni-Suef. In accordance to our results, many studies reported the high prevalence of *T. annulata* compared to other piroplasms in cattle from different provinces in Egypt [87–89]. In the current study, we observed that the majority of cases (10 out of 15) were detected in cattle from El-Wady El-Geded province that in accordance with Al-Hosary et al. [89], who stated that the prevalence of *T. annulata* in cattle from El-Wady El-Geded province was 63.6%. This finding might be due to the climate in this province, which is dry and sunny throughout the year, which is conducive to tick activity [89]. Likewise, we identified

*T. ovis* in sheep from Giza and Beni-Suef and buffaloes from Beni-Suef. In Egypt, there are few studies reporting *T. ovis* in sheep [90] and buffaloes [91]. In parallel, a recent study reported that *T. ovis* was detected in sheep from Menoufia and El-Wady El-Geded province [92], implying that this pathogen is widespread in sheep throughout Egypt. Finally, we detected one case of *Ba. canis* in a dog from Cairo province with 100% identity with *Ba. canis vogeli* detected in a dog from Egypt (GenBank: AY371197). Canine babesiosis is distributed worldwide and was later detected in Egypt by Passos et al. [93] and Salem and Farag [94]. In Africa, *Ba. canis vogeli* has been detected in different regions such as South Africa [95], Tunisia [96] and Côte d'Ivoire [80].

Family Anaplasmataceae was known to cause human and animal diseases, is transmitted by ticks and has a worldwide distribution [26,97]. In the current study, the overall prevalence of anaplasmosis was 30.9% (172/557) by qPCR, while we obtained only 48 samples with good quality sequences, possible due to the higher sensitivity of qPCR compared to standard PCR or due to the co-infection with family Anaplasmataceae. The overall infection rate of *An. marginale* was 3.8% (21/557) in cattle, sheep and camels from different localities (Beni-Suef, Qena, El-Wady El-Geded and Cairo). In Egypt, *An. marginale* was first mentioned in the national report in 1966, after which the disease was reported in numerous provinces [32–34,98]. Several studies reported endemicity of *An. marginale* in cattle [16,28,31–34], buffaloes [30] and camels [29]. However, *An. marginale* was detected for the first time in sheep. To our knowledge, *An. marginale* has not yet been described in sheep. For the first time, *An. centrale* was detected in a bovine from El-Wady El-Geded province, Egypt. *Anaplasma centrale* is closely related to *An. marginale* but less pathogenic, so it has been used as a live vaccine to protect against bovine anaplasmosis [99,100]. We also found that sheep and cattle from Beni-Suef province (upper Egypt) were positive for *An. ovis* with a prevalence rate of 0.7% (4/557). To the best of our knowledge, *An. ovis* has never been detected in cattle and sheep in Egypt. In parallel, a recent study reported that *An. ovis* was detected in sheep in Menoufia province (one of Delta provinces) [34], implying that this pathogen is widespread in cattle and sheep throughout Egypt. *Anaplasma ovis* is the etiological agent of ovine anaplasmosis in small ruminants and causes mild and subclinical infections [23]. In Africa, some studies reported *An. ovis* in sheep from Tunisia [101], Senegal [25] and Algeria [102,103], and in cattle from Algeria [103]. In addition, we found that dogs from Cairo and a camel from Giza province were positive for two genotypes of *An. platys*, with an infection rate of 1.4% (8/557). In Egypt, *An. platys* was never molecularly identified in dogs and camels. Later, Loftis et al, [51] detected *An. platys* in ticks collected from dogs. *Anaplasma platys* is the causative agent of canine anaplasmosis, which causes severe thrombocytopenia in dogs [104]. Interestingly, we detected that cattle, buffaloes and sheep from Beni-Suef province and camels from Giza and Cairo provinces were positive for a new potential *Anaplasma* sp. with a prevalence rate of 2.5% (14/557). This probably new species was genetically related to canine *An. platys*, which is why it was commonly referred to as *An. platys*-like. This *An. platys*-like genotype has never been detected in Egypt, except in a recent study where *An. platys*-like bacterium was detected only in cattle in Menoufia province [34], implying that this new potential pathogen circulates between different animal hosts (excluding dogs that seem to be susceptible for a type *An. platys* only) and different provinces in Egypt. Later, *An. platys*-like was detected in various animal hosts such as cattle in Italy [105], Algeria [106] and Tunisia [107], camels in Tunisia [108,109] and sheep and goats in South Africa [110] and Senegal [25]. Various *Anaplasma* sp. were identified by the 23S RNA gene and which further confirmed by the *rpoB* gene.

Rickettsioses are VBDs of humans and animals and are mainly transmitted by ticks [35]. In Africa, the human pathogens *R. africae*, *R. aeschlimannii*, *R. conorii* and *R. massiliae* have been identified in ticks and animals [39–41]. In our study, rickettsial DNA was detected in dogs

from Capital Cairo with a prevalence of 1.5% (3/203) in dogs. Phylogenetic analysis showed that our genotypes (*R. africae*-like) clustered in a separate and well-supported branch (bootstraps 94) with *R. africae* previously detected in Egypt (Fig 5) [53]. To the best of our knowledge, *R. africae* has not been previously detected in dogs anywhere in the world. Thus, this is the first detection of *R. africae*-like pathogens in dog anywhere in the world. African tick-bite fever, a benign disease with severe complications in elderly populations, and transmitted mainly in the south and West Africa by *Amblyomma variegatum* [35,111]. Likewise, *R. africae* was identified in other tick genera as *Hyalomma* sp. [42,53,54,112] and in *Rh. sanguineus* (the most common tick parasitizing dogs) [113].

Relapsing fever borrelioses caused by group of the spirochete group *Borrelia* sp. and is transmitted by soft and hard ticks [57]. In the present study, we identified *Bo. theileri* in bovine and ovine blood for the first time in Beni Suef province, Egypt, with an overall prevalence of 0.9% (5/557). Alignment of five sequences obtained revealed that there is a new potential genotype of *Bo. theileri* circulating between cattle and sheep in Beni-Suef province, which is 99% identical to *Bo. theileri* found in *Rh. geigyi* in Mali [58]. *Borrelia theileri* is considered one of the relapsing fever borreliae and the etiological agent of bovine borreliosis in cattle, transmitted by hard ticks, mainly *Rhipicephalus* sp. [114]. In Egypt, *Bo. theileri* was reported in *Rh. annulata* collected from donkeys in the same province [115]. Later, *Bo. theileri* was also detected in *Rh. annulata* in Egypt [62]. Recently, some studies have detected *Bo. theileri* in cattle such as Argentina [116] and Cameroon [117]. Similarly, *Bo. theileri* has been detected in the blood of sheep in Algeria [102]. It appears that, *Bo. theilerii* is not exclusively pathogenic to cattle.

Q fever is a tick-borne disease that is a major public health concern [65]. The infection in human manifests as acute or chronic febrile disease often associated with endocarditis and abortion [65]. In Egypt, Q fever was first detected in a high-risk group of cattle farmers [118]. Later, many reports demonstrated the prevalence of the disease in goats, sheep, cattle and camels [67–72,119,120]. In this study, the overall prevalence of Q fever in sheep and goats from Sinai province is 0.3% (3% in goats and 1.7% in sheep). This result was in accordance with Abdel-Moein and Hamza [71] who reported an overall prevalence of Q fever of 0.9% and 3.4% in goats. PCR and sequencing amplified only Cox2 with a 100% match with the *C. burnetii* reference recorded in GenBank, while genotyping and sequencing of the positive samples with other spacers (Cox5 & Cox18) failed. This result can be explained by the fact that the high sensitivity of qPCR can detect low DNA concentrations and the lower prevalence of *C. burnetii* in blood is lower than feces and urine [121,122].

For filarial infections, we detected four cases of filarial infection with an overall prevalence of 0.7%, 1.5% (3/203) in dogs and 0.7% (1/148) in camels. In dogs from the capital Cairo, we identified two different species of Filariidae as *D. repens* and *Acanthocheilonema* sp. *Acanthocheilonema viteae* is the filarial nematode of rodents, while *Ac. reconditum* is the etiological agent of filariasis in dogs. Also, there is no sequence of *Ac. reconditum* for the 18S rRNA gene in GenBank. Therefore, we suspect that the identified species is, however, *Ac. reconditum*. Therefore, this is the first report of *D. repens* and *Ac. reconditum* in dogs in Egypt. Subcutaneous dirofilariasis of domestic dogs is caused by *D. repens* and is common in Africa, Asia and Europe [123]. It is a mosquito-borne nematode that is a public health problem [124]. *Acanthocheilonema reconditum* colonizes the peritoneal cavity and adipose tissue and can cause skin lesions with allergy and is transmitted by fleas and biting lice [78,125,126]. In Africa, some studies reported microfilariae of *Ac. reconditum* in dogs in South Africa [127], Côte d'Ivoire [80]. Moreover, a camel from Giza province was positive for filarial nematodes, and was identified as *S. digitata*. To our knowledge, *S. digitata* has not been previously detected in camels. *Setaria digitata* is the natural filarial nematode of the Bovidae and the adult worm is resident

in the peritoneal cavity [128,129]. Accidental transmission of *S. digitata* to unnatural hosts such as horses, donkeys, sheep and goats causes worrisome pathological problems such as corneal opacity and blindness [74,130,131–133].

Finally, we reported 1.3% (7/557) co-infections in animals, with the highest percentage in cattle 5.7% (5/557). Co-infection in cattle is common and has been reported in many studies [33,34,117,134]. We observed that all cases of co-infections including *Anaplasma* sp. with another pathogen such as piroplasms, *Borrelia* or even *Rickettsia*. Regarding the endemicity of VBDs, we observed the most infected region in Beni-Suef province, where the same genotypes or even new potential pathogens circulated between different animal hosts with a risk of transmission to other adjacent provinces and to humans. Furthermore, we observed that the highest prevalence among animal hosts was anaplasmoses (48/557; 8.6%), followed by piroplasmoses (23/557; 4.1%). Molecular analysis revealed an interesting diversity of these VB pathogens in ruminants and dogs. Therefore, further studies are needed for a better understanding of the epidemiological mapping of pathogen-host-vector in this region or even in the whole Egypt.

In conclusion, the current study is the first large-scale epidemiological observational study that performed molecular screening and characterization of multiple vector-borne pathogens in different animal hosts for better understanding of the endemicity of VBDs in Egypt. We identified for the first time *An. centrale*, *An. ovis*, a new *An. platys*-like and *Bo. theileri* in cattle, a new *An. platys*-like in buffaloes, *An. marginale*, *An. ovis*, a new *An. platys*-like and *Bo. theileri* in sheep, *An. platys*, a new *An. platys*-like and *S. digitata* in camels and *R. africae*-like, *An. platys*, *D. repens* and *Ac. reconditum* in dogs in Egypt. Therefore, ruminants and dogs in Egypt are reservoirs for multiple neglected, emerging and re-emerging vector-borne pathogens, especially new potential pathogens. Our observational study aimed to describe the repertory of possible vector-borne zoonotic pathogens in Egypt. However, convenient sampling approach did not permit us to evaluate the association of identified pathogens with host characteristics and to describe the geographic distribution of pathogens that limited our study. Further studies are needed to determine the pathogen-host-vector connections and other epidemiological factors of VBDs throughout Egypt, as well as to decipher the zoonotic potential of newly identified genotypes and their animals and public health significance.

## Author Contributions

**Conceptualization:** Hend H. A. M. Abdullah, Oleg Mediannikov.

**Data curation:** Hend H. A. M. Abdullah.

**Formal analysis:** Hend H. A. M. Abdullah, Didier Raoult, Oleg Mediannikov.

**Funding acquisition:** Oleg Mediannikov.

**Investigation:** Hend H. A. M. Abdullah.

**Methodology:** Hend H. A. M. Abdullah, Nadia Amanzougaghene, Handi Dahmana, Meriem Louni, Oleg Mediannikov.

**Project administration:** Oleg Mediannikov.

**Resources:** Didier Raoult, Oleg Mediannikov.

**Software:** Hend H. A. M. Abdullah, Nadia Amanzougaghene, Oleg Mediannikov.

**Supervision:** Didier Raoult, Oleg Mediannikov.

**Validation:** Didier Raoult, Oleg Mediannikov.

**Visualization:** Didier Raoult, Oleg Mediannikov.

**Writing – original draft:** Hend H. A. M. Abdullah.

**Writing – review & editing:** Hend H. A. M. Abdullah, Nadia Amanzougaghene, Handi Dahmana, Meriem Louni, Didier Raoult, Oleg Mediannikov.

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
