## [Decision Letter · Decision Letter 0]

19 Mar 2021

Dear Dr. Mediannikov,

Thank you very much for submitting your manuscript "Multiple vectors borne diseases in Domestic Animals in Egypt" for consideration at PLOS Neglected Tropical Diseases. As with all papers reviewed by the journal, your manuscript was reviewed by members of the editorial board and by several independent reviewers. In light of the reviews (below this email), we would like to invite the resubmission of a significantly-revised version that takes into account the reviewers' comments. Please pay careful attention to each of the Reviewers' comments, as many relate to the clarity of presentation of the data and the validity of the conclusions. Correction of the english language errors will be important to the resubmission. 

One Reviewer stated that this work is not appropriate for this journal, so please strengthen the aspects of the work that focus on disease as well as on the presence of the pathogens, particularly since infection does not necessarily lead to disease.

We cannot make any decision about publication until we have seen the revised manuscript and your response to the reviewers' comments. Your revised manuscript is also likely to be sent to reviewers for further evaluation.

Sincerely,

Jenifer Coburn, PhD

Associate Editor

Amanda Bastos

Deputy Editor

Reviewer's Responses to Questions

**Key Review Criteria Required for Acceptance?**

**Methods**

-Are the objectives of the study clearly articulated with a clear testable hypothesis stated?

-Is the study design appropriate to address the stated objectives?

-Is the population clearly described and appropriate for the hypothesis being tested?

-Is the sample size sufficient to ensure adequate power to address the hypothesis being tested?

-Were correct statistical analysis used to support conclusions?

-Are there concerns about ethical or regulatory requirements being met?

Reviewer #1: Abstract

Line 13: you can just say vector-borne pathogens.

Line 21: please replace ‘proved’ with found.

Line 26: the presence of different….

Line 28: ruminants seem to play….

Author summary

Line 34: please remove ‘was’ after that.

Line 39: these findings suggest that….

Line 40: please rephrase the end part of the sentence to ‘emerging and re-emerging potentially new vector borne pathogens that have significant implications in human health’ or something along those lines.

Introduction

Line 46: humans, livestock, companion animals and…

Line 46: vbds are a worldwide burden….

Line 50: such as.

Line 51: spread of VBDs…

Line 51: such as globalization…

Line 53: pose.

Line 54 – 112: these paragraphs can be deleted or significantly condensed.

Line 118: this sentence needs to be rephrased.

Line 119: provide.

Materials and Methods

This section is fairly straightforward and concise but the other sections are heavily wordy and can be greatly condensed.

Line 128: I’m assuming all these animals were domesticated? Please mention it explicitly.

Line 161: ‘performed’ instead of applied.

Reviewer #2: The objectives of the study are articulated with the hypothesis. It is not clear what are the criteria for inclusion of animals in the research and how the sample size calculation was performed to obtain the prevalence. In addition, they mention that information such as gender, breed, age, health status and vector infestation were collected, but it is not described or analyzed in the manuscript. There are no concerns about complying with ethical or regulatory requirements.

Reviewer #3: The objectives of the study were clearly stated.

The study design did not appropriately address the stated objective

The population were not described and hence cannot be measured with the study hypothesis

The sample size is not sufficient in ensuring adequate power to address the aim off the study

More analysis needs to be carried out. Authors should mention exactly where sampling was done in the provinces whether households, kraals, abattoir and their respective confounders to the study.

**Results**

-Does the analysis presented match the analysis plan?

-Are the results clearly and completely presented?

-Are the figures (Tables, Images) of sufficient quality for clarity?

Reviewer #1: I think the whole results section needs to be reworked. As it is written, it is very difficult to read and grasp the main points of the research. The emphasis is heavily placed on the pathogens and not on the animals, but I think that it should be the other way around (this is not a microbiology journal). For example, instead of saying “No Bartonella sp. DNA was detected in all animal hosts, while, piroplasms, Anaplasma sp., Rickettsia sp., Borrelia sp., C. burnetii and Filaria sp. DNA were detected in different animal hosts (Table 3)” please write it as ‘None of the animals were positive for Bartonella sp. while different animal species were positive for……..’ or something along those lines. 

In line 133: You mention that you also recorded data about each animal’s gender, breed, age, vector infestation and health status but you do not report whether any of these factors had an effect on the detection of pathogens.

Reviewer #2: The results of the analysis are adequate and clearly described. The figures and tables were well chosen and have quality. I suggest the inclusion of a map of Egypt showing the origin of the animals.

Reviewer #3: The analysis matches the analysis plan, but more analysis needs to be done.

The results are clearly presented but not complete.

The figures (Tables, Images) are of sufficient quality for clarity

**Conclusions**

-Are the conclusions supported by the data presented?

-Are the limitations of analysis clearly described?

-Do the authors discuss how these data can be helpful to advance our understanding of the topic under study?

-Is public health relevance addressed?

Reviewer #1: Needs to be rewritten. I could not make sense of the results section. Therefore, I don’t think any of my specific comments on the discussion/conclusions section would be helpful. I would request the authors to please rewrite the results and discussion sections from mainly the host animals’ perspective and not from the pathogens’ perspective. Also, please try to be concise as the way it is written is very wordy.

I would like to see table 3 with the rows and columns switched.

Reviewer #2: The conclusion is adequate, but the authors do not describe the limitations of the analysis. The relevance of the results for advancing the discussion of the subject and the importance of the findings for public health are highlighted.

Reviewer #3: The conclusion are not supported by the data presented.

The limitations of the study were not mentioned

**Editorial and Data Presentation Modifications?**

Reviewer #1: This manuscript can be greatly condensed to better explain the results and their importance in the discussion sections.

Reviewer #2: Line 21: replace “Dirofilariarepens” by “Dirofilaria repens”

Line 55: replace “Piroplamida” by “Piroplasmida”

Lines 119-121: It is not the first data... There are at least two published works with objectives similar to the present manuscript:

Tumwebaze MA, et al. Parasitol Int. 2020 Oct;78:102150. doi: 10.1016/j.parint.2020.102150

AL‑Hosary et al. Parasites Vectors (2020) 13:495 https://doi.org/10.1186/s13071-020-04372-z

Lines 128-131: It is not clear what are the criteria for inclusion of animals in the research and how the sample size calculation was performed to obtain the prevalence. I suggest the inclusion of a map of Egypt showing the origin of the animals. 

Lines 132-133: The authors mention that information such as gender, breed, age, health status and vector infestation were collected, but these data are not described or analyzed in the manuscript. 

Lines 135-136: Why was the blood extraction kit not used?

Lines 149-151, and line 166: Was a reagent blanck control used?

Line 190: Was a reagent blanck control used?

Line 191: How was the sample size determined for the calculation of prevalence? What is the justification for using the term "Prevalence" throughout the text?

Lines 285-290: It is important to describe the presence of co-infections also in the abstract

Lines 318-319: Replace the term “E. canis” by “Ba. canis”. In addition, citation number 89 refers to the first molecular description of Babesia vogeli in Brazil, so it does not seem appropriate to use this reference here.

Lines 338-339, 356-357, 427-431: This study reports the first detection of A. ovis in sheep and A. platys-like strains in cattle in Menoufia and Egypt:

Tumwebaze MA, Lee SH, Adjou Moumouni PF, Mohammed-Geba K, Sheir SK, Galal-Khallaf A, Abd El Latif HM, Morsi DS, Bishr NM, Galon EM, Byamukama B, Liu M, Li J, Li Y, Ji S, Ringo AE, Rizk MA, Suzuki H, Ibrahim HM, Xuan X. First detection of Anaplasma ovis in sheep and Anaplasma platys-like variants from cattle in Menoufia governorate, Egypt. Parasitol Int. 2020 Oct;78:102150. doi: 10.1016/j.parint.2020.102150 

Lines 383-384: Borrelia theileri description in Brazil was from ticks, not from bovine samples

Line 397: Replace “faces” by “feces”

Lines 425-433: It is importante to describe the limitations of the analysis in conclusion.

Reviewer #3: (No Response)

**Summary and General Comments**

Reviewer #1: Overall, I think that the manuscript needs to be improved a lot more to be considered for reevaluation.

Reviewer #2: The present work demonstrates important data about agents transmitted by vectors in animals in Egypt, contributing to reduce an existing gap on this knowledge. 

There are some recent works describing some pathogens that the authors thought were unpublished data. However, this does not detract from the research that was really extensive, covering various genera of pathogens transmitted by vectors to farm and companion animals.

Reviewer #3: TITLE

‘Multiple vectors borne diseases in domestic animals in Egypt’... Title is misleading it was not the disease that was investigated but the presence of pathogens.

ABSTRACT

Abstract needs to be re-written. Certain sentences do not read well and need restructuring. Authors should check how capitals and small letters are used and rectify all anomalies. The type of study should be stated in the abstract. Line 26 – 27 states that ‘For the first time, we detected the presence different zoonotic, mostly vector-borne pathogens in the blood of domestic animals in Egypt…. The question is by the use of mostly is the author implying that some of the pathogens were not VBD? If this is so which ones are those and why include them in the write up since the aim of the study was on VBD. All the new genotype discovered during the study should be mentioned under the abstract section. 

INTRODUCTION

Some of the diseases mentioned such as piroplasmosis presents itself as asymptomatic in livestock, some pathogens like Theileria present itself in the asymptomatic and is not zoonotic therefore what is its economic importance to merit it being studied. In inference the mortality and morbidity of these diseases should be highlighted clearly so that the relevance of its pathogens being studied is put in proper perspective.

On line 46 what is the meaning of companion in that context.

METHODS

The section was not well written. Authors should rewrite it by arranging it systematically following the order of; study area or study setting description-make sure each province is discussed in detail including their population, seasons and topography etc, sampling design and procedure-should be as detailed as possible including how each animal was sampled per province, where e etc, exclusion criteria, statistical analysis, ethical approval and consent. 

The convenient sampling that was carried out does not allow for authors to generalize the study to the whole country. The authors should be clear on the distribution of the animals in the provinces in the country and why. Authors should stop beginning sentences with acronym. Describe how field and laboratory processes were carried out.

OVERALL 

For substantial portion of the article it is not sequentially written out to allow for flow of thought.

RESULTS

There should be another table that shows the demographics, vector infestation and health status of the animals with respective analysis. Reconcile from line 187 to that of the breakdown of the various VBD pathogens mentioned at the abstract section as well as with the various tables.

DISCUSSION

Line 312 and 319 should state the authors whose studies this research is referring to before the references are written in parenthesis.

Check tenses, missing words and misuse of words and correct them. 

Authors should also include in the discussion dynamics of all the VBD as present in the provinces, as per their proximity to each other and the potential beneficial or detrimental effect they might pose on the people of living in and around the study provinces and then the nation as a whole.

It should also be discussed the new strain found their potential virulence or pathogenecity. 

How all pathogens identified economically impact the general populace should be further discussed.

CONCLUSION

Based on the sampling technique the conclusion is farfetched, the convenient sampling done in this study cannot be used in projecting what pertains in the whole country

PLOS authors have the option to publish the peer review history of their article (what does this mean?). If published, this will include your full peer review and any attached files.

Reviewer #1: No

Reviewer #2: No

Reviewer #3: No
---

## [Decision Letter · Decision Letter 1]

23 Jun 2021

Dear Dr. Mediannikov,

Thank you very much for submitting your manuscript "Multiple Vectors-borne Pathogens in Domestic Animals in Egypt" for consideration at PLOS Neglected Tropical Diseases. As with all papers reviewed by the journal, your manuscript was reviewed by members of the editorial board and by several independent reviewers. The reviewers appreciated the attention to an important topic. Based on the reviews, we are likely to accept this manuscript for publication, providing that you modify the manuscript according to the review recommendations. Please pay careful attention to the points raised by Reviewer 3, and please make every effort to have your manuscript proofread for correct usage of the English language. For example, there is a grammatical error in the Title.

Sincerely,

Jenifer Coburn, PhD

Associate Editor

Amanda Bastos

Deputy Editor

Reviewer's Responses to Questions

**Key Review Criteria Required for Acceptance?**

**Methods**

-Are the objectives of the study clearly articulated with a clear testable hypothesis stated?

-Is the study design appropriate to address the stated objectives?

-Is the population clearly described and appropriate for the hypothesis being tested?

-Is the sample size sufficient to ensure adequate power to address the hypothesis being tested?

-Were correct statistical analysis used to support conclusions?

-Are there concerns about ethical or regulatory requirements being met?

Reviewer #2: The objectives of the study are related to the hypothesis. The authors clarified the criteria for inclusion of animals (convenience sampling) in the research, how the prevalence was calculated and added, as suggested, a figure containing the location of the samples obtained on the map of the country. 

The authors, when asked about not presenting results from data mentioned in the material and methods (about sex, race, age, health status and vector infestation), justified that these data will be presented on another article. Therefore, I suggest that this part is not mentioned as part of this manuscript. 

There are no concerns about complying with ethical or regulatory requirements.

Reviewer #3: Authors clearly articulated the objectives of the study with the right study design and sufficient sample size but the animal population was not clearly described with respect to their sex, breed, age, vector infestation and health as stated in line 135.

**Results**

-Does the analysis presented match the analysis plan?

-Are the results clearly and completely presented?

-Are the figures (Tables, Images) of sufficient quality for clarity?

Reviewer #2: Data results are adequate and clearly described. The figures and tables were well chosen and of good quality. The authors accepted the suggestion to include a map of Egypt with the location of the samples collected.

Reviewer #3: The analysis presented does match the analysis plan with the figures (Tables, Images) being of sufficient quality for clarity however authors on several occasions at the results section restated their methods adding up to the bulkiness of the manuscript. 

In line 209 how was 172 out of 557 arrived at for Anaplasmataceae in the animal hosts?

The table 3 shows An. platys in 7 dogs why is line 226 reporting 6?

Reconcile line 230 and 231 to table 3.

Include the An. platys-like pathogen obtained from two dogs in table 3.

**Conclusions**

-Are the conclusions supported by the data presented?

-Are the limitations of analysis clearly described?

-Do the authors discuss how these data can be helpful to advance our understanding of the topic under study?

-Is public health relevance addressed?

Reviewer #2: The conclusion is adequate and the authors pointed out the limitations of the analysis. The relevance of the results for advancing the discussion of the subject and the importance of the findings for public health are highlighted.

Reviewer #3: yes the conclusions are supported by the data presented with limitations adequately described

**Editorial and Data Presentation Modifications?**

Reviewer #2: The required changes were made, with no further changes suggested by my review.

Reviewer #3: Minor edits

Reword line 67, and lines 69 to 70 to convey appropriately the message it wishes to convey. 

Authors should check spacing and punctuations through out the manuscript.

Abbreviations of all scientific names should be synchronized through out the manuscript eg A. ovis and An. ovis, and A. platy and An. platy.

The prevalence of piroplasmoses have been stated on line 193 and 194, it needn't be repeated on line 197.

Sentence on lines 214 to 216 should be rewritten to convey the information it is intended to send to readers.

On line 424 replace ...' and there is'... with ...' with'...

**Summary and General Comments**

Reviewer #2: (No Response)

Reviewer #3: Authors should not repeat most of the results in the discussion section, just discuss.

Discussion on piroplasmoses with respect to T. ovis should be more detailed. 

Are all the organisms transmitted by ticks? If not kindly mention the other vectors and discuss them with their corresponding pathogens. 

Again all zoonotic pathogens should be mentioned as well as their devastating effect.

PLOS authors have the option to publish the peer review history of their article (what does this mean?). If published, this will include your full peer review and any attached files.

Reviewer #2: No

Reviewer #3: No

Figure Files:

Data Requirements:

Reproducibility:

References

---

## [Editor Report · Decision Letter 2]

26 Aug 2021

Dear Dr. Mediannikov,

We are pleased to inform you that your manuscript 'Multiple Vector-borne Pathogens of Domestic Animals in Egypt' has been provisionally accepted for publication in PLOS Neglected Tropical Diseases.

Best regards,

Jenifer Coburn, PhD

Associate Editor

Armanda Bastos

Deputy Editor

---

## [Editor Report · Acceptance letter]

14 Sep 2021

Dear Dr. Mediannikov,

We are delighted to inform you that your manuscript, "Multiple Vector-borne Pathogens of Domestic Animals in Egypt," has been formally accepted for publication in PLOS Neglected Tropical Diseases.

Best regards,

Shaden Kamhawi

co-Editor-in-Chief

Paul Brindley

co-Editor-in-Chief
